# Silver Nanoparticles Agglomerate Intracellularly Depending on the Stabilizing Agent: Implications for Nanomedicine Efficacy

**DOI:** 10.3390/nano10101953

**Published:** 2020-09-30

**Authors:** Marina R. Mulenos, Henry Lujan, Lauren R. Pitts, Christie M. Sayes

**Affiliations:** Department of Environmental Science, Baylor University, Waco, TX 76798-7266, USA; Marina_George@baylor.edu (M.R.M.); Henry_Lujan@baylor.edu (H.L.); Lauren_Pitts@baylor.edu (L.R.P.)

**Keywords:** silver nanoparticles, transformation, hyperspectral imaging, stabilizing agent, agglomeration, drug delivery, nanomedicine

## Abstract

Engineered nanoparticles are utilized as drug delivery carriers in modern medicine due to their high surface area and tailorable surface functionality. After in vivo administration, nanoparticles distribute and interact with biomolecules, such as polar proteins in serum, lipid membranes in cells, and high ionic conditions during digestion. Electrostatic forces and steric hindrances in a nanoparticle population are disturbed and particles agglomerate in biological fluids. Little is known about the stability of nanoparticles in relation to particle surface charge. Here, we compared three different surface-stabilized silver nanoparticles (50 nm) for intracellular agglomeration in human hepatocellular carcinoma cells (HepG2). Nanoparticles stabilized with branched polyethyleneimine conferred a positive surface charge, particles stabilized with lipoic acid conferred a negative surface charge, and particles stabilized with polyethylene glycol conferred a neutral surface charge. Particles were incubated in fetal bovine serum, simulated lung surfactant fluid, and simulated stomach digestion fluid. Each nanoparticle system was characterized via microscopic (transmission electron, fluorescence, and enhanced darkfield) and spectroscopic (hyperspectral, dynamic light scattering, and ultraviolet-visible absorption) techniques. Results showed that nanoparticle transformation included cellular internalization, agglomeration, and degradation and that these changes were dependent upon surface charge and incubation matrix. Hyperspectral analyses showed that positively charged silver nanoparticles red-shifted in spectral analysis after transformations, whereas negatively charged silver nanoparticles blue-shifted. Neutrally charged silver nanoparticles did not demonstrate significant spectral shifts. Spectral shifting indicates de-stabilization in particle suspension, which directly affects agglomeration intracellularly. These characteristics are translatable to critical quality attributes and can be exploited when developing nano-carriers for nanomedicine.

## 1. Introduction

Silver nanoparticles (AgNPs), first used in consumer health-related products in the late 1800s, continue to be incorporated into modern medicine due to their unique properties [1]. Properties such as small size and large surface area, relative low toxicity, and stability while in suspension are useful in drug delivery carriers (DDCs) [2,3]. However, information regarding agglomeration or degradation of AgNPs after in vivo administration is scarce. Many studies suggest that silver ions leached from AgNP surfaces induce significant toxicity [4,5,6,7]. While most studies focus on the effects of silver on microorganisms in the environment, only a few examine the effects of AgNPs in human physiologically relevant conditions [8,9,10]. Given the interest in using silver and other metal-based nanoparticles as potential DDCs, it is imperative to gain a deeper understanding of potential nanoparticle transformations in biological fluids.

Substantial attention has been given to the beginning and ending stages of nano-enabled drug product development pipelines [11,12]. Quality-by-design approaches in nanomaterial manufacturing, risk analyses, and hazard assessments are recommended to ensure safe and efficacious products. Physicochemical characterization (PCC) serves to identify in situ critical quality attributes (CQAs), enabling considerations of interactions between nano-enabled diagnostics, therapeutics, or theragnostics with different physiological fluids, such as cerebral, blood and serum, lung surfactant, interstitial, mucous, breast milk, digestive, bile, and urine [13,14]. Due to the high variability of nanomaterials used in pharmaceutical research and development, CQAs are needed to improve and streamline synthesis and production methods, identify administration routes of exposure, prove efficacy in drug testing, and ensure biocompatibility during the ADME process (adsorption, distribution, metabolism, and excretion) [15,16]. Currently, the most critical attributes of nano-enabled medicines are size homogeneity after production, consistent active ingredient concentration, and stability of product over time [17,18]. Similar characteristics are applicable to other DDCs, such as liposomes, dendrimers, polymeric micelles, and other microspheres [19,20,21].

Currently, one of the most extensively studied surface coatings used in nanomedicine research and development is polyethylene glycol (PEG). Other relevant surface stabilizing agents include branched polyethyleneimine (bPEI), citrate, ascorbic acid, and oleic acid. bPEI is a common cationic coating and electro-steric stabilizing agent used in drug products [22]. Studies using environmental models have found that bPEI coatings induce nanoparticle bioaccumulation and teratogenicity in aquatic organisms [23,24,25]. Citrate and ascorbic acid stabilized nanoparticles have been proposed for use in dentistry, as these stabilizing agents have been shown to eradicate biofilm formation, while being a less toxic option relative to other types of antibiotics [26]. Citrate, ascorbic acid, and oleic acid coatings create a charge barrier preventing intracellular uptake when compared to bPEI or PEG [27]. Still, PEG is often the preferred choice for nanoparticle stability in nanomedicine formulations because this agent protects particles from reticuloendothelial system elimination [28].

Greater focus is given to AgNP transformation in environmentally relevant conditions as compared to AgNPs subjected to physiologically relevant conditions. Of the limited information available for AgNP characteristics in biofluids, most data focus on the nanoparticle protein corona construct [29,30,31]. Protein corona studies are relevant to nanoparticle transformations administered through intramuscular, subcutaneous, intravenous, or intradermal injection. However, other routes of nanomedicine exposure introduce additional types of complex nanoparticle interactions, transformations, agglomeration, and degradation [32]. Other administration routes include oral, nasal, inhalation, ocular, and transmucosal, including buccal, vaginal, rectal, and transdermal [33]. When nanomedicine exposure occurs orally, nanoparticles are subjected to saliva and stomach acid digestion. Through nasal, ocular, and transmucosal administrations, nanoparticles are subjected to mucosa. Through inhalation, nanoparticles are subjected to lung surfactant fluid. Through transdermal routes, nanoparticles are subjected to sebum.

Given that nanoparticles are currently used in drug delivery systems, the goal of this study was to compare the effects of surface stabilizing agents, and induced surface charge, on nanoparticle physiologically relevant transformation, including cellular internalization, agglomeration, and degradation. We utilize hyperspectral imaging coupled with a battery of analytical techniques and human hepatocellular carcinoma (HepG2) cells for this evaluation. We transformed three different AgNPs (50 nm); nanoparticles stabilized with branched polyethyleneimine (bPEI) conferred a positive surface charge, nanoparticles stabilized with lipoic acid conferred a negative surface charge, and nanoparticles stabilized with polyethylene glycol (PEG) conferred a neutral surface charge. Each nanoparticle system was incubated with polar proteins from serum, lipids from surfactant fluid, and hydrochloric acid from gastric fluid. This work outlines a reproducible methodology that can be repeated when characterizing nano-enabled drug product transformations after administration. These characteristics are translatable to critical quality attributes and can be exploited to develop complex in vitro, in vivo, or in silico models for nanomedicine and associated risk analyses.

## 2. Materials and Methods

### 2.1. Silver Nanoparticles

All nanoparticles used in this study were purchased from NanoComposix (San Diego, CA, USA) at a concentration of 1 mg/mL. Per manufacturer’s certificate of analysis, the three AgNPs included (1) branched polyethyleneimine polymer (1 mg/mL) at 25 kDa bound with primary amines to the silver nanoparticle surface (termed Pos–AgNP), (2) carboxyl lipoic acid (1 mg/mL) covalently bound with a dithiol to the silver nanoparticle surface (termed Neg–AgNP), and (3) methoxy-polyethylene glycol-coated silver nanoparticles (1 mg/mL) (termed Neu–AgNP).

### 2.2. Incubation Scenarios

Aliquots of AgNPs were incubated in three different scenarios to simulate protein adsorption, lipid interaction, and stomach acid dissolution. For Scenario 1, i.e., protein absorption simulating intravenous injection, AgNPs were separately incubated in USDA-approved fetal bovine serum (FBS; Gibco, Thermo Fischer Scientific, Waltham, MA, USA). For Scenario 2, i.e., lipid interaction simulating inhalation administration into surfactant fluid, AgNPs were incubated with a mixture of ovine cholesterol derived lipids (Avanti Polar Lipids, Alabaster, AL, USA) and didodecyldimethylammonium bromide surfactant solution (Sigma-Aldrich, St. Louis, MO, USA). For Scenario 3, i.e., stomach acid incubation simulating oral administration, AgNPs were incubated in hydrochloric acid supplemented with porcine stomach enzymes (Thermo Fischer Scientific, Waltham, MA, USA). All nanoparticle simulations were performed at 37 °C after a 24 h post-exposure time period. The concentration of silver in all resultant samples was 0.5 mg/mL, unless otherwise indicated.

### 2.3. Dynamic Light Scattering

Hydrodynamic diameter, dispersity index, and zeta potential analyses were performed via dynamic light scattering (DLS) spectroscopy with a Zetasizer Nano ZS spectrometer (Malvern Panalytical, Malvern, United Kingdom) at 25 °C (Appendix A). All measurements were performed in a 1060-folded capillary zeta cell (Malvern Panalytical, Malvern, United Kingdom) at a 1:1000 dilution in ultrapure water (18 Ohms, MilliQ Ultrapure water purification system, GenPure-Thermo Fischer Scientific, Waltham, MA, USA). Hydrodynamic diameter and dispersity index analyses were performed with 11 replicates per sample and run in triplicate; zeta potential measurements were performed with 50 replicates per sample and run in triplicate. Measurements were taken before and after AgNP transformation.

### 2.4. Transmission Electron Microscopy

Nanoparticle morphology was assessed by transmission electron microscopy (TEM; JEM-1010, JEOL Inc., Akishima, Tokyo, Japan). AgNPs were collected at 24 h post-incubation and deposited on a copper formvar-coated grid (EMS, Hatfield, PA, USA) for 5 min. Once dried, loaded grids were imaged via TEM with a spot size of 2.0 and an acceleration voltage of 60 kV. ImageJ software was used for image analyses.

### 2.5. Hepatocyte Cell Culture

Human hepatocellular carcinoma (HepG2) cells were cultured in Eagle’s minimum essential medium (EMEM; Gibco, Thermo Fisher Scientific, Waltham, MA, Unites States) supplemented with 10% fetal bovine serum (FBS; Equitech-Bio, Inc. Kerrville, TX, USA) and a 1% penicillin/streptomycin mixture (MP Biomedical, Solon, OH, USA). Cells were cultured in an air-jacked humidified incubator at 37 °C with 5% CO_2_. Once cells were grown to 70% confluency, they were inoculated with a 1 μg/mL suspension of transformed AgNP systems for 24 h.

### 2.6. Fluorescence Imaging

Cellular morphology was assessed via fluorescence imaging. HepG2 cells were plated into chamber slides coated for adherent cells (Nunc, Lab-Tek Chamber Slide, Thermo Scientific, Waltham, MA, USA) and grown to 70% confluency. Cells were inoculated with a 1 μg/mL suspension of AgNP systems for 24 h and fixed with the Image-iT Fixation/Permeabilization kit (Thermo Fisher Scientific, Waltham, MA, USA). Fluorescent dyes included MitoTracker Red CM-H2-XRos (579/599 nm) to visualize reactive oxygen species, NucBlue LiveReadyProbes (360/430 nm) to visualize the nucleus, and ActinGreen 488 ReadyProbes (495/518 nm) Reagent (AlexaFluor 488 phalloidin) to visualize the cytoskeleton. To preserve the slides, ProLong Diamond Antifade Mountant was used. All cell slide preparation items were purchased from Thermo Fischer (Waltham, MA, USA).

Subsequently, the preserved cells were imaged using a CytoViva^®^ Fluorescent microscope (Auburn, AL, USA) with fluorescent excitation cubes for DAPI (4′,6-diamidino-2-phenylindole), FITC (fluorescein isothiocyanate), and TRITC (tetramethylrhodamine 5-isothiocyanate chloride) with a 40X oil-immersion lens.

### 2.7. Hyperspectral Imaging

Samples were prepared as stated in Section 2.6. Enhanced darkfield images were acquired using a 40X lens within the CytoViva^®^ Hyperspectral System. Exposure time of 0.25 s was used, with high spatial resolution and a 2.5 nm spectral resolution. The field of view was selected via positioning from the fluorescence images. Intensities of each sample were between 2000 and 3500 to ensure sufficient spectra data. Once images were collected, they were enhanced using the linear 2% function to visualize the nanoparticles intracellularly. Spectral libraries were created with nanoparticles that were taken up into cells and spectra were averaged from 30 selected regions of interest (ROI). Data were normalized against cell and background corrections. Samples were created with nanoparticles before incubation in physiological scenarios as well as after inoculation to HepG2 cells.

### 2.8. Inductively Coupled Plasma-Mass Spectrometry

Samples were analyzed for dissolution as a function of time using inductively coupled plasma-mass spectrometry (ICP-MS) (Appendix A). Samples were subjected to simulated incubations for 1, 24, and 48 h. Silver ions were isolated from the nanoparticles and surrounding solution through centrifugation. Once centrifuged, an aliquot of the supernatant was collected for acid digestion. The metal isotope calibration standard (SPEX CertiPrep, Thermo Fischer Scientific, Waltham, MA, USA), sample aliquots, method blank (consisting of each transformation solution mentioned above), and an acid blank were placed in digestion tubes and a 1:4 mixture of hydrochloric to nitric acids was added. A watch glass was placed over top and the digestion was prepared for 2 h at 95 °C in a heat block. Hydrochloric acid, nitric acid, plastic digestion tubes, and plastic watch glasses were TraceMetalGrade and purchased from Thermo Fischer Scientific (Waltham, MA, USA). After digestion, samples were filtered (0.2 μm) and diluted with ultrapure water to 2%. An internal standard mix (Agilent Technologies, Santa Clara, CA, USA) was used for drift correction. All measurements were acquired on an Agilent ICP-MS 7900 from Agilent Technologies (Santa Clara, CA, USA). Runs were performed in triplicate with isotopes ^107^Ag and ^109^Ag monitored. All data analysis was performed using MassHunter software (Agilent Technologies, Santa Clara, CA, USA).

### 2.9. Ultraviolet-Visible Absorption Spectroscopy

Ultraviolet-visible (UV-Vis) absorption spectroscopy was used to assess nanoparticle degradation over time using a Lambda 35 photospectrometer (Perkin Elmer, Waltham, MA, USA) (Appendix A). Data were analyzed on UV WinLab software (PerkinElmer, Waltham, MA, USA). Ultrapure water was used as a blank and dilutant producing a 1:2 dilution of the original sample for analysis. In a quartz cuvette, scans were obtained from 200 to 800 nm in three separate sessions during each time point (1, 24, and 48 h) in triplicate.

### 2.10. Statistical Analyses

To perform statistical analysis, standard deviation and standard error were calculated with Excel (Microsoft, Redmond, WA, USA). Spectral averaging and normalization were completed with ENVI software (Cytoviva, Auburn, AL, USA). Spectra were plotted with GraphPad Prism (San Diego, CA, USA).

## 3. Results

### 3.1. Experimental Design

In this study, three different biological fluids, relevant to nanomedicine administration routes, were produced and used to study potential silver nanoparticle (AgNP) transformations. Incubation of AgNPs with fetal bovine serum, ovine cholesterol-derived lipids, and simulated gastric fluid are referred to as Scenario 1, 2, and 3, respectively. Scenario 1 represents nanoparticles administered intravenously and interactions with serum proteins. Scenario 2 represents nanoparticle inhalation and interactions with interstitial fluid. Scenario 3 represents oral administration and interactions with stomach acid. For each scenario, nanoparticle suspensions were incubated for 24 h at 37 °C. Figure 1 outlines the experimental details, inclusive of nanoparticle characterization, incubation parameters, and endpoint analyses. The branched polyethyleneimine-stabilized silver nanoparticles are termed “Pos–AgNPs”; the lipoic acid-stabilized silver nanoparticles are termed “Neg–AgNPs”; and the methoxy polyethylene glycol-stabilized silver nanoparticles are termed “Neu–AgNPs”. A comprehensive physicochemical characterization was completed prior to transformations to determine concentration and enable comparisons post-incubation (Appendix A).

Simulations of Scenario 1, i.e., nanoparticle interactions with polar biomolecules within the circulatory system, have been previously performed [30,34,35,36,37]. Data show that nanoparticle size, surface charge, and stabilizing agents influence protein corona formation (Appendix A). Nanoparticle protein corona is the self-assembly of absorbed proteins on the particle surface. The nanoparticle surface and size determine the amount, identity, polarity, and arrangement of the protein corona, while also influencing downstream effects of distribution, metabolism, and excretion [29,30,34]. Depending on the biomedical application, the formation of a protein corona could either be efficacious or produce an ineffective biological response [35,36]. Previous studies show that the protein corona can act as a cloaking shield during targeted drug delivery; a pre-coating of proteins may reduce the particle–protein interactions in vitro. In contrast, the development of the protein corona on the surface of the nanomedicine may render the therapeutic agent detrimental by routing the medicine to undesired targets [37].

Scenario 2 represents interactions between nanoparticles and lung interstitial fluid after inoculation and incubation; the available information from the published literature indicates that nanoparticle surfaces can be exploited to promote lipid raft transport across membranes [38,39]. Interactions with lipids induce changes in surface properties as nanoparticles pass through cytoplasmic, mitochondrial, or nuclear membranes. Kang et al. (2008a and 2008b) showed that positively charged AgNPs facilitate stable transport across olefin/paraffin separation membranes [40,41]. However, once internalized by the cell, the metal-based nanoparticles are likely to be shuttled into lysosomes, where ions dissociate and induce cellular toxicity. Recent efforts have been aimed at preparing nanoparticle surface coatings to shuttle active pharmaceutical ingredients into the cytoplasmic membrane and avoid cellular internalization. Identifying the surface charge and coating that partially interacts with lipid membranes while avoiding intracellular acidification aids in reducing adverse effects of nano-enabled drug carriers [42].

Scenario 3 represents ingestion and digestion of orally administered nanoparticles. Most studies focus on AgNPs used in food products [43]. Cueva et al. (2019) showed that nanoparticles in simulated digestion fluid are structurally different post-digestion but do not alter the microbiota viability [44]. Other studies have shown that citrate-coated nanoparticle aggregation is pH-dependent in the gastrointestinal tract. At pH of 2, nanoparticles form > 100 nm aggregates; at pH of 5, nanoparticles aggregate less and often degrade [45].

### 3.2. Engineered Nanoparticles for Nano-Enabled Drug Products

To understand the potential transformations of AgNPs under the three different scenarios used in this study, physicochemical characterization of the nanoparticles was performed before incubation. Figure 2 highlights key characteristics of the pre-incubated nanoparticles. In suspension, Pos–AgNP was positively charged (+ 73 mV), as measured by zeta potential, whereas Neg–AgNP and Neu–AgNP were both negatively charged, with zeta potential values of −57 and −23 mV, respectively. It is generally regarded that nanoparticle surface charge, as measured by zeta potential, greater than + 30 mV or less than −30 mV indicates a stable nanoparticle suspension [46]. As seen in Appendix A, the primary factor that influenced the dissolution of the nanoparticles was incubation media, not surface charge. The neutrally charged silver nanoparticles have steric stabilization mechanisms, which is not as strong as the other two samples, which have electrostatic and electrosteric mechanisms, thus increasing the potential for complete dissolution.

Neu–AgNP, with a zeta potential measurement of −23 mV, indicates an unstable suspension. This observation is confirmed via transmission electron microscopy (TEM), where small nanoparticle aggregates are clearly present. Pos–AgNP and Neg–AgNP zeta potential values indicate stable suspensions. TEM analyses of pre-incubated Pos–AgNPs show slight agglomeration. Agglomeration could be due to drying effects produced during sample preparation [47,48]. All three pre-incubated AgNP samples are spherical in shape and ~50 nm in diameter.

Nanoparticles (and nanoparticle agglomerates) below 20 nm in diameter are rapidly cleared by renal excretion. Nanoparticles (and nanoparticle agglomerates) larger than 200 nm in diameter are easily cleared by the reticuloendothelial system [49,50]. For these reasons, among others, nanomedicine research and development efforts have favored ~50 nm nanoparticle diameters to prolong systemic circulation and prevent unintended elimination.

### 3.3. Hyperspectral Imaging for Intracellular Agglomeration Analysis

Figure 3 shows data from enhanced darkfield hyperspectral imaging (HSI) for HepG2 cells exposed to nanoparticles. Darkfield images were collected using the Environment for Visualizing Images (ENVI) software. Corrections were performed to eliminate interferences from cell membranes, background, and lamp spectrum. A spectral library was created and only included the nanoparticles that were internalized within cells; nanoparticles on the cell surface or in the background were excluded. Regions of interest (ROIs) were obtained over an averaged spectrum of >25 nanoparticles per sample. Data were normalized and compared to pre-incubated AgNPs, where the averaged spectra are plotted in Figure 3.

Pos–AgNPs (Figure 3A,G) showed a spectral shift to the right (i.e., an increase in wavelength), while the Neg–AgNPs (Figure 3B,H) showed a spectral shift to the left (i.e., a decrease in wavelength). Neu–AgNPs showed differential spectral shifts depending on the incubation scenario. Regarding Scenario 1, a blue shift was observed (Figure 3C), in contrast to Scenario 3, where a red shift was observed (Figure 3I). After subjection to Scenario 2, AgNPs exhibited no shift in hyperspectral analysis (Figure 3F). In fact, none of the AgNPs subjected to the lipid surfactant simulation (i.e., Scenario 2) showed a shift in hyperspectral analysis (Figure 3D–F). Interestingly, Scenario 2 did not have any effect on nanoparticle internalization, probably due to severe agglomeration (>500 nm).

Longer wavelengths of detected light (i.e., red shifts) observed in HSI are associated with an increase in nanoparticle size or, in this case, an increase in agglomeration. Shorter wavelengths (i.e., blue shifts) are indicative of a decrease in nanoparticle size, such as dissolution or de-aggregation, or partial degradation [51,52]. We show that Pos–AgNPs agglomerate intracellularly, whereas Neg–AgNPs and Neu–AgNPs tend to produce mixtures of single nanoparticles and small agglomerated nanoparticles independent of the scenario. The transformation mechanism could involve the incomplete degradation of the AgNP surface stabilizing agent or the partial dissolution of silver cations (Ag^+^) from the nanoparticle surface. Ion dissolution was confirmed with inductively coupled plasma-mass spectrometry (ICP-MS) analyses (Appendix A). Either transformation mechanism could produce an increase in reactive oxygen species (ROS), oxidative stress, or phosphorylation of proteins, lipids, and enzymes [53]. AgNP surface charge and coating play roles in defining safety and efficacy critical attributes [2,3].

### 3.4. Biotransformed Silver Nanoparticles Used in Drug Products

Figure 4 shows AgNPs subjected to Scenario 1 (i.e., serum incubation). Dispersity index (DI, a measure of polydispersity within the sample) was minimal (0.110, 0.335, and 0.154 for Pos–AgNPs, Neg–AgNPs, and Neu–AgNPs, respectively). However, the hydrodynamic diameter was large (102.8, 313.1, and 88.6 nm for Pos–AgNPs, Neg–AgNPs, and Neu–AgNPs, respectively). These data suggest that nanoparticles transformed into severely agglomerated entities. Agglomeration, in this scenario, is due to an abundance of proteins adsorbed onto the surface of each AgNP, subsequently creating a protein corona hindering nanoparticle stability by masking the stabilizing agent, as indicated by the similar zeta potential measurements (−25.2, −20.4, and −20.2 mV for Pos–AgNPs, Neg–AgNPs, and Neu–AgNPs, respectively). Neg–AgNPs had the largest change in DI, hydrodynamic diameter (HDD), and surface charge (as measured by zeta potential) compared to the Pos–AgNP and Neu–AgNP systems. This comparison may be the result of partial agglomeration and incomplete protein corona formation seen with TEM imaging (Figure 4C–E).

All three AgNP systems underwent a significant biotransformation under Scenario 1. Agglomeration in this scenario occurs both extracellularly, i.e., when nanoparticles first interact with serum proteins, as well as intracellularly as indicated by hyperspectral imaging. Human hepatocellular carcinoma cells (HepG2) were used for the intracellular analyses. When HepG2 cells were exposed to AgNP subjected to Scenario 1, no significant change in cell morphology or biomarker expression was observed (Figure 4F–H). This indicates that nanoparticle agglomeration may not induce unintended cellular effects but could deactivate any therapeutic activity presented by the nanoparticle carrier administered intravenously.

Figure 5 shows AgNPs subjected to Scenario 2 (i.e., lipid surfactant incubation). In this scenario, Neg–AgNPs changed the most in DI, HDD, and zeta potential measurements compared to the Pos–AgNP and Neu–AgNP systems. The Pos–AgNP and Neu–AgNP systems became unstable, as indicated by a change in zeta potential from their pre-incubated charges to unstable values of −25.43 ± 0.59 mV and −20.73 ± 1.06 mV, respectively. Neg–AgNPs, in contrast to the two other nanoparticles, had a stable zeta potential measurement at −31.40 ± 0.70mV. The Neg–AgNPs also agglomerated significantly, with an HDD of 461.63 ± 61.57 nm, which is three times and seven times greater than the size of either Pos–AgNP or Neu–AgNP systems. Neu–AgNP exhibited a lower DI and only changed −7 nm in HDD (i.e., 59.74 ± 0.64 nm), indicating subtle nanoparticle transformation. When examining the effects of Scenario 2 transformed AgNP systems on the HepG2 cells, Neg–AgNPs induced a significant amount of oxidative stress in the cells when compared to either Pos–AgNP or Neu–AgNP systems (Figure 5F–H). However, cytoskeletal degradation was observed in HepG2 cells exposed to Pos–AgNP in this scenario (Figure 5F). No change was reported for the Neu–AgNP sample in the HepG2 system. Nanoparticles release ions in solution which may interact with the cell surface and disrupt actin networks. Some nanoparticles disrupted the actin network more than others due to the method of diffusion into the cell. Cytoskeleton heterogeneity may play a role in nanoparticle uptake.

Figure 6 shows AgNPs subjected to Scenario 3 (i.e., stomach acid incubation). Like Scenario 1, the Pos–AgNP, Neg–AgNP, and Neu–AgNP systems became unstable, as indicated by a change in zeta potential from their pre-incubated charges to unstable values of –16.63, –28.07, and –17.23, respectively. In this scenario, Neg–AgNPs and Pos–AgNPs agglomerated, whereas Neu–AgNPs decreased in HDD. This observation could be due to the increased likelihood of charged particles interacting with surrounding biomolecules and the high ionic concentration present in the scenario. In short, Neu–AgNPs simply degraded in the acidic environment. When examining the effects of Scenario 3 transformed AgNP systems on the HepG2 cells, Pos–AgNPs induced a significant amount of oxidative stress and cytoskeleton damage in the cells when compared to either Neg–AgNP or Neu–AgNP systems (Figure 6F–H).

Taken together, Pos–AgNP, Neg–AgNP, and Neu–AgNP systems transformed under all scenario conditions. The transformation products can be related to the pre-incubation zeta potential measurements. These results show that neutrally charged nanoparticles do not react with polar biomolecules or high ionic content, causing little to no particle transformation. Not only are strongly surface-charged nanoparticles able to enter cells more readily, but they also interact with biomolecular constituents in the complex microenvironment and accumulate intracellularly differently [54]. Thus, cellular effects induced by these particle types are likely to be caused by the increased oxidative stress, which is consistent with what has been extensively studied with AgNP behavior in the environment [55]. Neutrally charged nanoparticle surfaces readily dissolve in acidic conditions. Therefore, effects induced by these particle types are more likely to be caused by metal ion dissociation and subsequent ionic interferences with normal cellular processes.

### 3.5. Negatively-Charged Silver Nanoparticles Influence Nanoparticle Biotransformation

Neg–AgNPs subjected to incubation in all three scenarios used in this study produced significantly transformed products in terms of a large shift in dispersity index, increased hydrodynamic diameter, weakened zeta potential, severe agglomeration, and induced cytoskeletal damage. Neu–AgNPs were not transformed as much as Neg–AgNPs due to the neutral surface being ineffective at interacting with the surrounding environment.

Pos–AgNP transformations were more noticeable than the Neu–AgNP results but were still minor when compared to those of Neg–AgNPs. Evidence of this was observed in the blue shift in Neg–AgNPs hyperspectral imaging. A blue shift is indicative of the nanoparticle de-aggregation or degradation, indicating that nano-enabled drug products with positively charged surfaces have more surface area available to interact with the complex mixtures. In comparison to the Pos–AgNPs had red shifts, indicating severe agglomeration, a decrease in surface area, and decrease in biomolecule interactions. The Neu–AgNPs having a slight red shift in Scenario 1, no shift in Scenario 2, and slight blue shift in Scenario 3 is indicative of neutrality and these particle transformations are largely influenced by the surrounding environment. These results indicate that there is a charge-dependent nanoparticle biotransformation mechanism at play. Agglomeration, cellular internalization, and degradation transformations of nano-enabled drug products may increase efficacy by transporting through the surrounding environment easily. Alternatively, transformation products may decrease effectiveness by modifying flux between biological compartments. Nanoparticle surface stabilizing agents can impact the therapeutic efficacy of nano-enabled drug products by not only altering stability but also dictating transformation after administration. The mechanisms of transformations observed in this study are summarized in Figure 7.

## 4. Conclusions

In summary, we studied three uniquely stabilized silver nanoparticles, each with a different surface charge and subjected to three separate physiologically relevant incubation scenarios. The surface charges included positive (conferred by bPEI), negative (conferred by lipoic acid), and neutral (conferred by PEG). The incubation scenarios included serum which represented intravenous administration, lipid surfactant fluid representing inhalation administration, and stomach acid representing oral administration. It is clear that AgNP system transformation products after subjection to each incubation scenario are strikingly different from their pre-incubated engineered structures. Changes in nanoparticle size and agglomeration (measured by hydrodynamic diameter), surface charge (measured by zeta potential), and agglomeration (measured by hyperspectral imaging and transmission electron microscopy) are evident and ought to be considered in nanomedicine research and development. In this study, we showed that the surface charge plays a substantial role in predicting agglomeration and intercellular uptake. A one size fits all characterization approach for nan-enabled drug products is not sufficient; multiple physicochemical properties are translatable to critical quality attributes.

## Figures and Tables

**Figure 1 nanomaterials-10-01953-f001:**
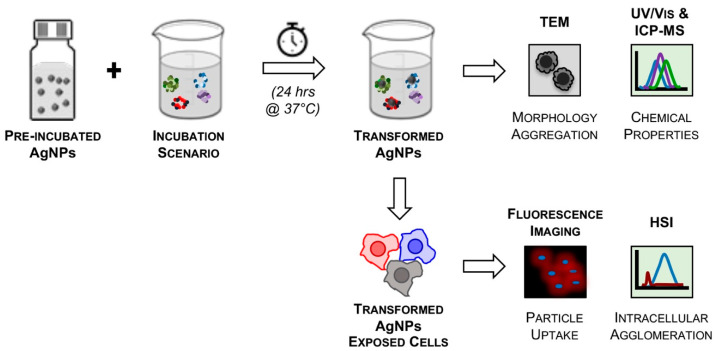
The experimental design used in this study. First, positive, negative, and neutral surface charged AgNPs were suspended in ultrapure water in separate stock suspensions. Second, each of the AgNP systems was subjected to one of three incubation scenarios for 1, 24, and 48 h. Third, analyses of transformed nanoparticles were performed using TEM (transmission electron microscopy) and HSI (hyperspectral imaging), among other techniques.

**Figure 2 nanomaterials-10-01953-f002:**
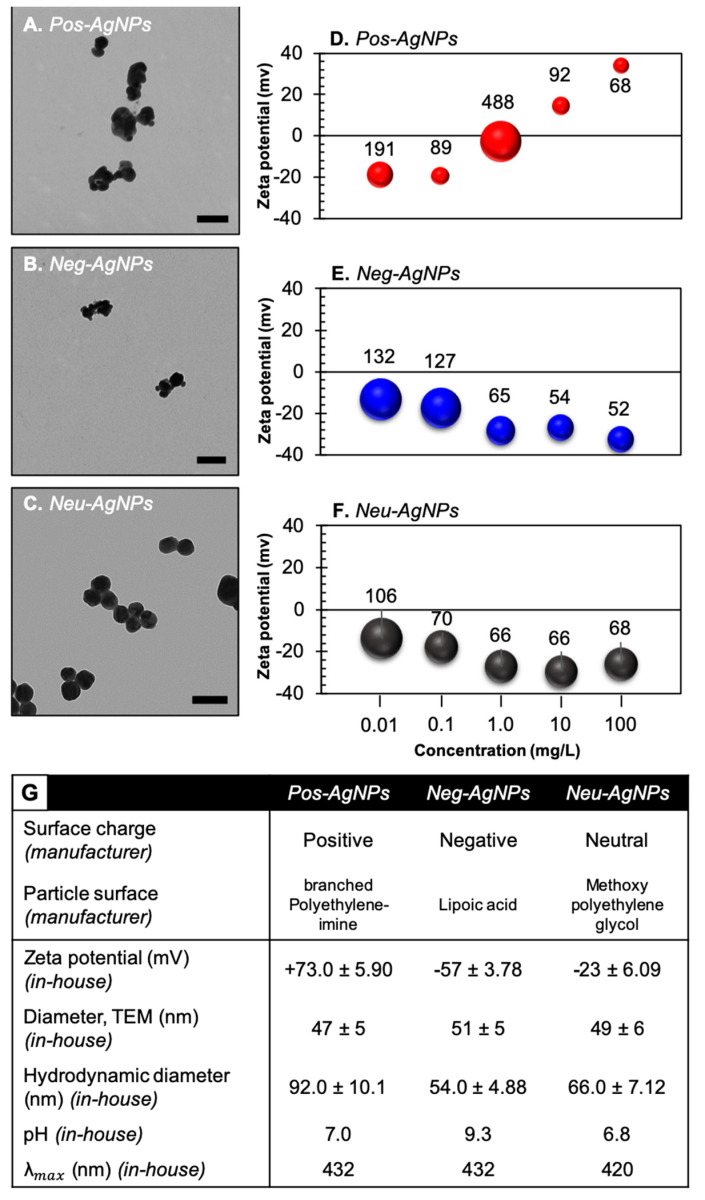
AgNP physicochemical characterization before incubation. Representative TEM images of (**A**) Pos–AgNPs, (**B**) Neg–AgNPs, and (**C**) Neu–AgNPs used in this study. Scale bar indicates 50 nm. Changes in zeta potential measurements (indicated by data point) and hydrodynamic diameter (indicated by value above data point) over increasing nanoparticle concentrations in ultrapure water for (**D**) Pos–AgNPs, (**E**) Neg–AgNPs, and (**F**) Neu–AgNPs used in this study. (**G**) Table of properties measured in-house or provided by nanoparticle manufacturer.

**Figure 3 nanomaterials-10-01953-f003:**
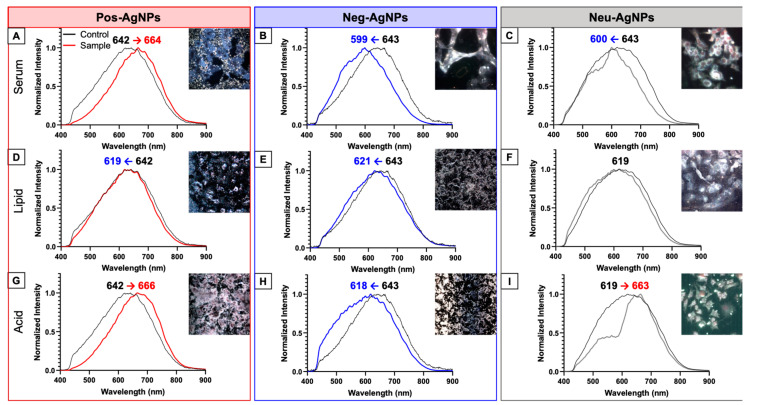
Enhanced darkfield and hyperspectral imaging of AgNPs subjected to incubation scenarios for 24 h and AgNP-exposed HepG2 cells for an additional 24 h. (**A**) Pos–AgNPs, (**B**) Neg–AgNPs, and (**C**) Neu–AgNPs incubated in serum. (**D**) Pos–AgNPs, (**E**) Neg–AgNPs, and (**F**) Neu–AgNPs incubated in lipids. (**G**) Pos–AgNPs, (**H**) Neg–AgNPs, and (**I**) Neu–AgNPs incubated in acid. The number noted above each spectrum indicates pre-incubated λ_max_ (black font) and post-incubated λ_max_ (blue or red font).

**Figure 4 nanomaterials-10-01953-f004:**
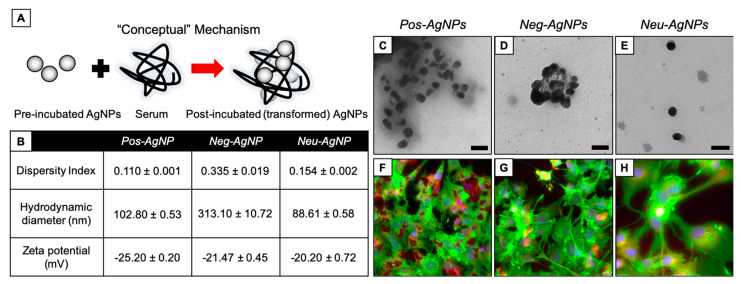
AgNP transformation after incubation in serum. (**A**) Conceptual model of transformation mechanisms. (**B**) Dispersity index (unitless), hydrodynamic diameter (nm), and zeta potential (mV) measurements of each particle type after simulated incubation. TEM of (**C**) Pos–AgNPs, (**D**) Neg–AgNPs, and (**E**) Neu–AgNPs after simulated incubation. Fluorescence microscopy of HepG2 cells after 24 h exposure to post-incubated (**F**) Pos–AgNPs, (**G**) Neg–AgNPs, and (**H**) Neu–AgNPs. Scale bars in (**C**–**E**) represent 50 nm. Stains in (**F**–**H**) include DAPI (blue), F-actin (green), and MitoTracker (red).

**Figure 5 nanomaterials-10-01953-f005:**
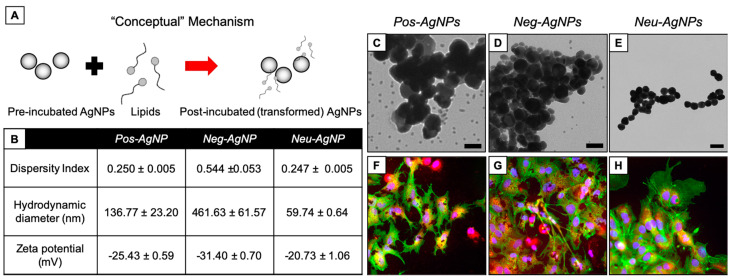
AgNP transformation after incubation in lipids. (**A**) Conceptual model of transformation mechanisms. (**B**) Dispersity index (unitless), hydrodynamic diameter (nm), and zeta potential (mV) measurements of each particle type after simulated incubation. TEM of (**C**) Pos–AgNPs, (**D**) Neg–AgNPs, and (**E**) Neu–AgNPs after simulated incubation. Fluorescence microscopy of HepG2 cells after 24 h exposure to post-incubated (**F**) Pos–AgNPs, (**G**) Neg–AgNPs, and (**H**) Neu–AgNPs. Scale bars in (**C**–**E**) represent 50 nm. Stains in (**F**–**H**) include DAPI (blue), F-actin (green), and MitoTracker (red).

**Figure 6 nanomaterials-10-01953-f006:**
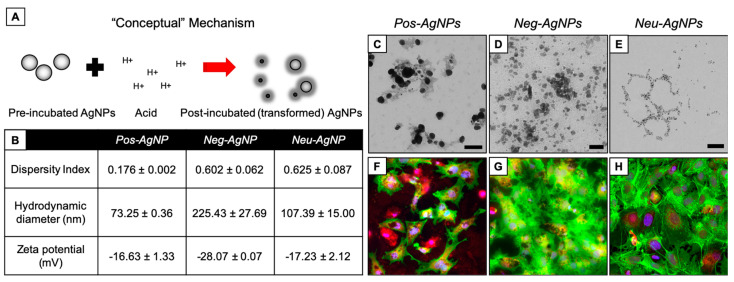
AgNP transformation after incubation in acid. (**A**) Conceptual model of transformation mechanisms. (**B**) Dispersity index (unitless), hydrodynamic diameter (nm), and zeta potential (mV) measurements of each particle type after simulated incubation. TEM of (**C**) Pos–AgNPs, (**D**) Neg–AgNPs, and (**E**) Neu–AgNPs after simulated incubation. Fluorescence microscopy of HepG2 cells after 24 h exposure to post-incubated (**F**) Pos—AgNPs, (**G**) Neg–AgNPs, and (**H**) Neu–AgNPs. Scale bars in (**C**–**E**) represent 50 nm. Stains in (**F**–**H**) include DAPI (blue), F-actin (green), and MitoTracker (red).

**Figure 7 nanomaterials-10-01953-f007:**
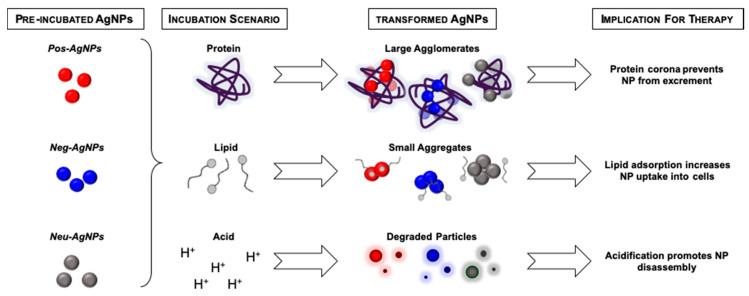
Summary of transformation mechanisms revealed by this study.

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
