# Peer review of "Silver Nanoparticles Agglomerate Intracellularly Depending on the Stabilizing Agent: Implications for Nanomedicine Efficacy"

_nanomaterials, 2020, doi:10.3390/nano10101953_

Round 1
Reviewer 1 Report
The work by Mulenos et al. is well planned and the experimental part is properly performed. However, the manuscript is, sometimes, difficult to read and a proper discussion is missing. Text revision (some typos are also present) and a discussion about feasible stabilizing agents suggested for different applications in biomedicine could improve the readability and the quality of the paper.
Author Response
Review Comments Summary
Reviewer 1
The work by Mulenos et al. is well planned and the experimental part is properly performed. However, the manuscript is, sometimes, difficult to read and a proper discussion is missing. Text revision (some typos are also present) and a discussion about feasible stabilizing agents suggested for different applications in biomedicine could improve the readability and the quality of the paper.
Response: We thank the reviewer for these helpful comments. The revised paper has been thoroughly reviewed for typos and grammatical errors.
A further discussion on the feasibility of stabilizing agents as related to different biomedical applications has been edited on page 2, “Currently, one of the most extensively studied surface coating used in nanomedicine research and development is polyethylene glycol (PEG). Other relevant surface stabilizing agents include branched polyethyleneimine (bPEI), citrate, ascorbic acid, and oleic acid. bPEI is a common cationic coating and electro-steric stabilizing agent used in drug products [22]. Studies using environmental models have found that bPEI coatings induce nanoparticle bioaccumulation and teratogenicity in aquatic organisms [23-25]. Citrate and ascorbic acid stabilized nanoparticles have proposed use in dentistry, as these stabilizing agents have been shown to eradicate biofilm formation, while being a less toxic option relative to other types of antibiotics [26]. Citrate, ascorbic acid, and oleic acid coatings create a charge barrier preventing intracellular uptake when compared to bPEI or PEG [27]. Still, PEG is often the preferred choice for nanoparticle stability in nanomedicine formulations because this agent protects particles from reticuloendothelial system elimination [28].”
A new sentence has been added on page 13, “Nanoparticle surface stabilizing agents can impact the therapeutic efficacy of nano-enabled drug products by not only altering stability, but also dictating transformation after administration.”.

Reviewer 2 Report
The main objective of the work is the comprehensive characterization and comparison of three different surface-stabilized (positively, negatively and neutrally charged) silver nanoparticles before and after internalization in human hepatocellular carcinoma cells.
As stated by the authors, three different scenarios, relevant to nanomedicine administration routes, were previously investigated; in fact AgNPs were characterized (before and after this step) and incubated with fetal bovine serum, ovine cholesterol-derived lipids, and simulated gastric fluid before exposition to HepG2 cells.
It is shown that AgNP system transformation products after subjection to each incubation scenario are strikingly different than their pre-incubated engineered structure. Changes in nanoparticle size and agglomeration (measured by hydrodynamic diameter), surface charge (measured by zeta potential), and agglomeration (measured by hyperspectral imaging and transmission electron microscopy) are evident and ought to be considered in nanomedicine research and development.
In my opinion, the quality of the manuscript is high in terms of process explanations and characterizations performed, average in terms of innovation and readibility.
Here some comments:
1) Figure 2: respect to the values discussed below the figure, in my opinion the scale bar of zeta potential for positive and negative AgNPs is wrong.
2) Referring again to Figure 2: why hydrodynamic diameter of pos AgNPs is higher than the other ones? Is there any explanation?
3) General comment: in my opinion, it would be nice to have further explanations regarding cytoskeletal degradation (when present) and decreasing of NPs zeta potential (and thus stability).
Why, Neg-AgNPs subjected to incubation in all three scenarios produced significantly transformed product respect to positive and neutral ones? It is evident from the characterization performed, but there’s a lack in the explanation.
4) Fig. 4-5-6. In the table, it would be probably useful to add TEM diameter to DI, HDD and zeta potential measurements.
5) In scenario 3, why Neu-AgNPs simply degraded in the acidic environment? It is not really clear to me.
6) Supplemental Figures and and their evidences should be further integrated in the discussion of the obtained results.
7) Spelling mistakes in line 38 (suspesnion) and line 300 (de-aggergation).
Author Response
Reviewer 2
The main objective of the work is the comprehensive characterization and comparison of three different surface-stabilized (positively, negatively and neutrally charged) silver nanoparticles before and after internalization in human hepatocellular carcinoma cells. As stated by the authors, three different scenarios, relevant to nanomedicine administration routes, were previously investigated; in fact, AgNPs were characterized (before and after this step) and incubated with fetal bovine serum, ovine cholesterol-derived lipids, and simulated gastric fluid before exposition to HepG2 cells. It is shown that AgNP system transformation products after subjection to each incubation scenario are strikingly different than their pre-incubated engineered structure. Changes in nanoparticle size and agglomeration (measured by hydrodynamic diameter), surface charge (measured by zeta potential), and agglomeration (measured by hyperspectral imaging and transmission electron microscopy) are evident and ought to be considered in nanomedicine research and development. In my opinion, the quality of the manuscript is high in terms of process explanations and characterizations performed, average in terms of innovation and readability.
Response: Thank you for these comments. And we appreciate the time it took for the reviewer to read our manuscript. We have made significant improvements in the readability of the text to help increase its impact.
Here some comments:
- Figure 2: Respect to the values discussed below the figure, in my opinion the scale bar of zeta potential for positive and negative AgNPs is wrong.
Response: we thank the reviewer for their comment and understand the concern. The data included in the table is the data we collected from the instrument. We suspect that when nanoparticles are suspended in solution, surface charge changes because of interactions between polar water molecules and highly charged nanoparticle surfaces. These interactions can influence (and change) the overall zeta potential measured, as seen in the data set.
- Referring again to Figure 2: Why hydrodynamic diameter of pos AgNPs is higher than the other ones? Is there any explanation?
Response: Our interpretation to this particular result is that positively charged nanoparticles tend to severely agglomerate in aqueous media; higher agglomeration usually results in higher hydrodynamic diameter measurements.
- General comment: in my opinion, it would be nice to have further explanations regarding cytoskeletal degradation (when present) and decreasing of NPs zeta potential (and thus stability).
Response: We thank the reviewer for this suggestion. On page 11, we added the following text, “Nanoparticles release ions in solution which may interact with the cell surface and disrupt actin networks. Some nanoparticles disrupted the actin network more than others due to the method of diffusion into the cell. Cytoskeleton heterogeneity may play a role in nanoparticle uptake.”
When nanoparticles agglomerate intracellularly, stability decreases because they are not interactions occur as agglomerates of nanoparticles as opposed to single particle interactions. (Grady ME, Parrish E, Caporizzo MA, Seeger SC, Composto RJ, Eckmann DM. Intracellular nanoparticle dynamics affected by cytoskeletal integrity. Soft Matter. 2017;13(9):1873-80.)
- Why, Neg-AgNPs subjected to incubation in all three scenarios produced significantly transformed product respect to positive and neutral ones? It is evident from the characterization performed, but there’s a lack in the explanation.
Response: We thank the reviewer for this suggestion. On page 13, we added the following text, “These results show that neutral charged nanoparticles do not react with polar biomolecules or high ionic content, causing little to no particle transformation.”
As seen in electron microscopy analyses, positively charged nanoparticles agglomerate and have significantly less surface area than the negatively charged nanoparticles. Bioavailable surface area corresponds directly with increased interaction with the surrounding environment.
- 4-5-6. In the table, it would be probably useful to add TEM diameter to DI, HDD and zeta potential measurements.
Response: This is an excellent suggestion and we will add these values to the next publication.
- In scenario 3, why Neu-AgNPs simply degraded in the acidic environment? It is not really clear to me.
Response: According to the recent literature, neutrally charged silver nanoparticles undergo oxidative dissolution in water and in ionic environments. This has been reported upon in environmental chemistry journals, i.e. Ag+ is produced when consumer products containing silver nanoparticles are released into the environment. (Radwan IM, Gitipour A, Potter PM, Dionysiou DD, Al-Abed SR. Dissolution of silver nanoparticles in colloidal consumer products: effects of particle size and capping agent. Journal of Nanoparticle Research. 2019 Jul 1;21(7):155.)
As seen in supplemental figure 1, the primary factor that influenced dissolution of the nanoparticles was incubation media, not surface charge. The neutrally charged silver nanoparticles have steric stabilization mechanisms, which is not as strong as the other two samples which have electrostatic and electrosteric, thus increasing the potential for a complete dissolution.
- Supplemental Figures and their evidences should be further integrated in the discussion of the obtained results.
Response: We thank the reviewer for this comment. On page 7, we added the following new text, “As seen in supplemental figure 1, the primary factor that influenced dissolution of the nanoparticles was incubation media, not surface charge. The neutrally charged silver nanoparticles have steric stabilization mechanisms, which is not as strong as the other two samples which have electrostatic and electrosteric, thus increasing the potential for a complete dissolution.”
We also corrected the spelling mistakes in line 38 (suspension) and line 300 (de-aggregation).

Reviewer 3 Report
Reviewing of the paper “Silver nanoparticles agglomerate intracellularly depending on the stabilizing agent: Implications for nanomedicine efficacy”
This manuscript describes the behaviour of silver nanoparticles treated with different stabilizing agents (branched polyethyleneimine, lipoic acid and polyethylene glycol) and their intracellular agglomeration profile in human hepatocellular carcinoma cells (HepG2).
Results showed that different nanoparticle stabilization agents affect cellular internalization, agglomeration and degradation and these changes are surface charge-dependent.
The originality of the work and the scientific relevance can be considered of a good level, considering that silver nanoparticles are widely used in consumer health-related products and to prepare modern nanodrugs.
The manuscript appears to be well organized, with good acknowledgement of the work of others in the references even if some minor enhancement can be added to text and pictures.
I think this paper is Acceptable with minor revisions.
Please add the following changes to the manuscript:
Page 9, line 343, “Biotransformed Silver Nanoparticles used in Drug Products Chapter”: the authors write: “Neg-AgNPs, in contrast to the two other nanoparticles, had a stable zeta potential measurement at -31.40 ± 0.70mV. The Neg-AgNPs also agglomerated significantly with an HDD of 461.63 ± 61.57. nm, which is 3X and 7X greater than the size of either Pos-AgNP or Neu-AgNP systems.” But authors also state at page 6, line 252: “It is generally regarded that nanoparticle surface charge, as measured by zeta potential, greater than +30 mV or less than -30 mV, indicates a stable nanoparticle suspension [47]. Therefore, any neutrally charged nanoparticle system (between -30 and +30 mV) is automatically considered to be unstable.”.
According this statement Neg-AgNPs have to be considered as stable nanoparticles, having a Z-potential lower than -30, even considering their standard deviation, but they are not, as it can easily seen from their hydrodynamic diameter and the TEM image present in Picture 5.
Can you add some more comments along the text? What is the reason according your opinion for this unconventional behaviour?
Supplemental Figure 3: I can’t find any Intensity value on the Y axes of all the 6 UV-Visible spectra herein present. I cannot understand if the curves are comparable or not among the different plots. Can you please add the Intensity values to all 6 plots?
Minor editing of English language and style are required, because I found typo and grammar errors during the text. I strongly suggest to revise the entire document from this point of view.
Only some of the errors I found are reported here below as examples:
- page 5 line 202: “…and interactions with serum proetins. Scenario…”;
- page 5 line 203: "…and intercations with interstitial fluid. Scenario…";
- page 8 line 290: “(i.e. an decrease in 290 wavelength)”;
- page 9 line 339: “…Figure 5. shows AgNPs…”, Figure 5 is written in Bold;
- page 9-10 line 345-346: “…agglomerated significantly with an HDD of 461.63 ± 61.57. nm, which is 3X and 7X greater than the size of either Pos-AgNP or Neu-AgNP systems…”;
- page 11 line 377: “…and Neu-AgNP systems ] transformed under all…";
- page 11 line 407: “…in this study are summarized in Figure 7. [Insert Figure 7 here

Author Response
Reviewer 3
This manuscript describes the behavior of silver nanoparticles treated with different stabilizing agents (branched polyethyleneimine, lipoic acid and polyethylene glycol) and their intracellular agglomeration profile in human hepatocellular carcinoma cells (HepG2). Results showed that different nanoparticle stabilization agents affect cellular internalization, agglomeration and degradation and these changes are surface charge-dependent. The originality of the work and the scientific relevance can be considered of a good level, considering that silver nanoparticles are widely used in consumer health-related products and to prepare modern nanodrugs. The manuscript appears to be well organized, with good acknowledgement of the work of others in the references even if some minor enhancement can be added to text and pictures. I think this paper is Acceptable with minor revisions.
Response: We thank the reviewer for their thorough review of our manuscript.
Please add the following changes to the manuscript:
Page 9, line 343, “Biotransformed Silver Nanoparticles used in Drug Products Chapter”: the authors write: “Neg-AgNPs, in contrast to the two other nanoparticles, had a stable zeta potential measurement at -31.40 ± 0.70mV. The Neg-AgNPs also agglomerated significantly with an HDD of 461.63 ± 61.57 nm, which is 3X and 7X greater than the size of either Pos-AgNP or Neu-AgNP systems.” But authors also state at page 6, line 252: “It is generally regarded that nanoparticle surface charge, as measured by zeta potential, greater than +30 mV or less than -30 mV, indicates a stable nanoparticle suspension [47]. Therefore, any neutrally charged nanoparticle system (between -30 and +30 mV) is automatically considered to be unstable”. According this statement Neg-AgNPs have to be considered as stable nanoparticles, having a Z-potential lower than -30, even considering their standard deviation, but they are not, as it can be easily seen from their hydrodynamic diameter and the TEM image present in Picture 5.
Response: We thank the reviewer for pointing this out. We have removed the following sentence from the manuscript, “Therefore, any neutrally charged nanoparticle system (between -30 and +30 mV) is automatically considered to be unstable.”
Can you add some more comments along the text? What is the reason according your opinion for this unconventional behavior?
Response: Zeta potential is one physicochemical characterization measurement that can indicate stability, typically with the surrounding environment as water. Here, negatively charged nanoparticles are considered stable due to the zeta potential measurement. But, as the reviewer points out, this is not always the case and often it depends on the surrounding conditions with polar biomolecules, lipids, or high ionic content. The surrounding environment is the cause of the ‘unconventional behavior’, which is a main point of this work, i.e. to show that stabilizing agents (as well as the surrounding matrix) influence nanoparticle transformation, and that nanoparticles must be studied in relevant conditions to help decipher mechanistic information.
We have added the following new text on page 7 of the paper, “As seen in Supplemental Figure 1, the primary factor that influenced dissolution of the nanoparticles was incubation media, not surface charge. The neutrally charged silver nanoparticles have steric stabilization mechanisms, which is not as strong as the other two samples which have electrostatic and electrosteric, thus increasing the potential for a complete dissolution.”
Supplemental Figure 3: I can’t find any Intensity value on the Y axes of all the 6 UV-Visible spectra herein present. I cannot understand if the curves are comparable or not among the different plots. Can you please add the Intensity values to all 6 plots?
Response: Graphs are of normalized intensities to visualize lambda max for all samples. Due to ionic dissolution, the concentration of all samples are not alike, therefore some samples do not show trends as easily. This has been fixed with normalization.
Minor editing of English language and style are required, because I found typo and grammar errors during the text. I strongly suggest to revise the entire document from this point of view.
Response: We thank the reviewer for these helpful comments. The revised paper has been thoroughly reviewed for typos and grammatical errors.
